# METHODS OF IMPROVING LLM TRAINING STABILITY

## ABSTRACT

Training stability of large language models (LLMs) is an important research topic. Reproducing training instabilities can be costly, so we use a small language model with 830M parameters and experiment with higher learning rates to force models to diverge, as in Wortsman et al. (2024). One of the sources of training instability is the growth of logits in attention layers Dehghani et al. (2023). We extend the focus of the previous work [Dehghani et al. (2023),Wortsman et al. (2024)] and look not only at the magnitude of the logits but at all outputs of linear layers in the Transformer block. We observe that with a high learning rate the L2 norm of all linear layer outputs grow with each training step and the model diverges. Specifically we observe that *QKV*, *Proj* and *FC2* layers have the largest growth of the output magnitude. This prompts us to explore several options: 1) apply layer normalization not only after *QK* layers (as it is done in [Dehghani et al. (2023), Wortsman et al. (2024)]) but after *Proj* and *FC2* layers too; 2) apply layer normalization after the QKV layer (and remove pre normalization). 3) apply QK layer normalization together with softmax capping. We show that with the last two methods we can increase learning rate by 1.5x (without model divergence) in comparison to an approach based on *QK* layer normalization only Dehghani et al. (2023). Also we observe significant perplexity improvements for all three methods in comparison to the baseline model.

## 1 INTRODUCTION

Research of transformer models training stability has gained a lot of attention in recent years [Chowdhery et al. (2024); Dehghani et al. (2023); Zhang et al. (2022); Touvron et al. (2021); Molybog et al. (2023); Wortsman et al. (2023); Wortsman et al. (2024); Zhai* et al. (2023)]. Multiple methods are proposed to improve model training stability. In Chowdhery et al. (2024), authors mitigate the issue by restarting training from a checkpoint roughly 100 steps before the spike in the loss started, and skip roughly 200–500 data batches. In Zhang et al. (2022) authors eliminate a subset of data as they found it increased the risk of instabilities. They also lower gradient clipping from 1.0 to 0.3. Training instability can be addressed by focusing on gradients with an optimizer [Molybog et al. (2023); Wortsman et al. (2023)]. AdamW-Adafactor is recommended in Wortsman et al. (2023)] to reduce loss spikes. One of the reasons for training instability is growth of logits in attention layers [Zhai* et al. (2023); Dehghani et al. (2023)]. So [Henry et al. (2020), Dehghani et al. (2023)] propose layer normalization after Q and K layers in the transformer block. Touvron et al. (2021) add learnable feature scaling after each residual block. That improves the training dynamic, allowing them to train deeper high-capacity image transformers. Another method of improving training stability is based on $\sigma$Reparam Zhai* et al. (2023), which reparametrize the weights of a linear layer. Towards the end of LLM training, output logits can diverge from the log probabilities Chowdhery et al. (2024). So Chowdhery et al. (2024) propose to use additional loss: *z_loss* to encourage the softmax normalizer to be close to 0. They found it improves the training stability. Weight decay also can be used to address model training divergence Wortsman et al. (2024).

One of the problems with exploring LLM training stability is the cost and time needed to reproduce the issue. In Wortsman et al. (2024), the authors propose training a small language model with a high learning rate to force the model to diverge early and simplify the analysis of training stability. We use the same approach in this work. As in [Zhai* et al. (2023); Dehghani et al. (2023); Wortsman et al. (2024)] we are focused on growth of logits in attention layers, but in addition we extend their work and explore the growth of the outputs of all linear layers in the transformer block.

- We extend the previous work Wortsman et al. (2024) and analyze not only the magnitude of the logits but all outputs of linear layers in the Transformer block. We show that during divergence the L2 norm of output layers *QKV*, *Proj* and *FC2* grow more than 2x in comparison to a converging model. This prompts us to explore several methods of improving model training stability: 1) apply layer normalization not only after *QK* layers (as it is done in [Dehghani et al. (2023), Wortsman et al. (2024)]) but after *Proj* and *FC2* layers too; 2) apply layer normalization after the QKV layer (and remove pre normalization); 3) apply QK layer normalization together with softmax capping.

- We show that two methods 1) layer normalization after QKV layer (without pre normalization); 2) combination of QK layer normalization together with softmax capping, allow to increase learning rate by 1.5x (without model divergence) in comparison to a method based on *QK* layer normalization only Dehghani et al. (2023). We also did thorough comparison of these approaches with multiple baseline methods of improving LLM training stability: $\sigma$Reparam, *soft_temp*, *soft_cap*, *soft_clip*, *LayerScale* and *QK_norm*.

- We show significant perplexity improvements (in comparison to the baseline model) with four methods explored in this paper: 1) QK layer normalization *QK* 2) apply layer normalization not only after *QK* layers (as it is done in [Dehghani et al. (2023), Wortsman et al. (2024)]) but after *Proj* and *FC2* layers too; 3) apply layer normalization after QKV layer (and remove pre normalization); 4) apply QK layer normalization together with softmax capping.

## 2 Experimental setup

We train a small version of an LLM with a similar experimental set-up as GPT-2 Radford et al. (2019) implemented in Megatron Shoeybi et al. (2020). The model has 830M parameters with 24 transformer blocks (topology of a one transformer block is shown on Figure 1.) It has a hidden size 1024 with 16 attention heads. The model is trained on a subset of a 1T token dataset with batch size 512 and sequence length 4096 using 32 H100 GPUs H10. We use an Adam optimizer Kingma & Ba (2015) with $\beta1 = 0.9$, $\beta2 = 0.95$ and gradient clipping at global norm 1. For learning rate, we use a linear schedule for warmup (where number of warmup samples is 122071) and a cosine decay schedule Loshchilov & Hutter (2017) for the remaining samples. We do not use weight tying of the embedding and output layer. We use rotary positional embeddings Su et al. (2024). We use a weight decay of 1e-1 and no dropouts. All linear layers (*QKV*, *Proj*, *FC1*, *FC2* on Figure 1) process data with bfloat16 precision. Weights in the optimizer are kept in float32 precision. Training data is a mixture of diverse set of public and proprietary datasets. The dataset contains 53 human languages and 37 programming languages. We use the SentencePiece tokenizer Kudo & Richardson (2018) to process text data.

One transformer block is shown on Figure 1. It has a *Multihead attention* module followed by a *Feed Forward* block (both have residual connections). The input sequence is processed by layer normalization(*LN*) followed by linear layer *QKV*. It projects features to *Q*, *K* and *V* for self attention computation in batched matmul (BMM1), Softmax and batched matmul(BMM2) layers. Its output is projected by linear layer *Proj*. *Feed Forward* block has layer normalization followed by fully connected layer *FC1*, activation function *SquaredReLU* So et al. (2021) and fully connected layer *FC2*. None of the linear layers (*QKV*, *Proj*, *FC1* and *FC2*) have additive bias. We label this model as bf16 baseline.

## 3 Model divergence analysis

As in Wortsman et al. (2024) we observe that with learning rate increase the probability of model divergence grows too. For example on Figure 2 we show loss function with learning rate 6e-3 (model converges, blue curve) and with learning rate 8e-3 (model diverges, black curve).

Transformer block (shown on Figure 1) has several linear layers: *QKV*, *Proj*, *FC1* and *FC2*. Linear layer takes input *X*, multiplies it with weights *W* and produces output *Y*. We use the above diverged and converged models (their loss functions shown on Figure 2) and plot their statistics depending on the training step in Table 1. We show how L2 norm of *W*, *X*, *Y* changes with every training step for *QKV*, *Proj*, *FC1* and *FC2* layers. While we only show statistics from the 2nd Transformer block

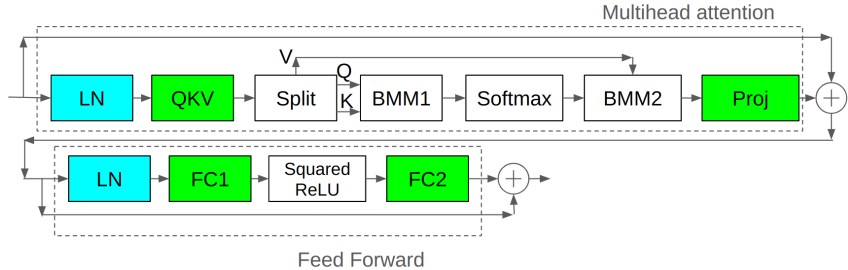

Figure 1: Transformer Block of bf16 baseline model.

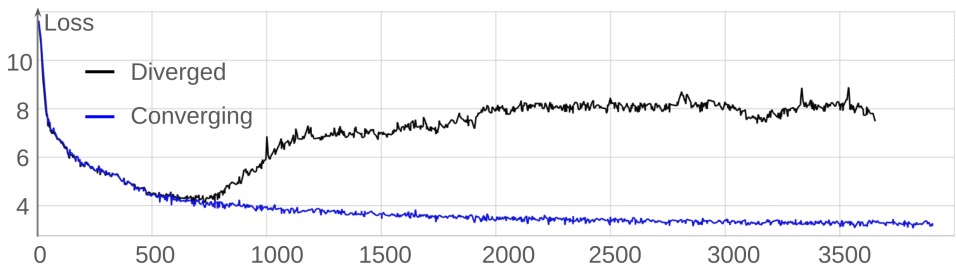

Figure 2: Example of training loss divergence/convergence (depending on training step).

here, the other Transformer blocks in the model had similar results. In Table 1 we see that with an increase of learning rate, L2 norm of $X$ and $Y$ grow a lot when the model diverges. In Table 2 we show input gradient explosion in *QKV* and *FC1* layers. Note that the input gradient of *FC2* and *Proj* layers in the diverged model look similar to the converged one (no gradient explosion, so we do not present it).

In Table 1 we show that the *QKV* layer has much higher magnitude output in divergent model in comparison to the converging one (e.g. L2 norm is more than 3x higher at training step 4000). This can create an issue with the softmax: its output can become almost one hot encoding, as reported in [Zhai* et al. (2023); Dehghani et al. (2023); Wortsman et al. (2024)] For demonstration purposes, let's generate logits $x$ from uniform distribution in range [-0.5...0.5] and show the output of softmax on Figure 3 (vertical axis is softmax output value, horizontal axis is an index of vector $x$, which has 16 values). We see that with increase of magnitude of $x$, output of softmax nears one hot encoding: for example if magnitude of $x$ is increased by 10x then almost all values become zero except 5 local maximas (blue curve on Figure 3). If magnitude of $x$ is increased by 40x then the output of softmax becomes one hot encoding with only one non zero value (red dashed curve on Figure 3). This kind of output in the softmax of the transformer can create issues with gradient propagation (as shown on Table 2) and loss divergence during training (as shown on Figure 2 and also reported in [Dehghani et al. (2023), Wortsman et al. (2024)].

## 4 METHODS OF IMPROVING MODEL TRAINING STABILITY

There are multiple options of dealing with Transformer training stability [Chowdhery et al. (2024); Dehghani et al. (2023); Zhang et al. (2022); Touvron et al. (2021); Molybog et al. (2023); Wortsman et al. (2023); Wortsman et al. (2024); Zhai* et al. (2023)]. This work is focused on improving training stability by controlling the magnitude of linear layers outputs, including logits.

### 4.1 σREPARAM

σReparam Zhai* et al. (2023) is a method to reparameterize the weights of a linear layer with:

$$\widehat{W} = \frac{\gamma}{\sigma(W)}W, \tag{1}$$

Table 1: L2 norm of *W*, *X*, *Y* for *QKV*, *Proj*, *FC1* and *FC2* layers (depending on training iteration)

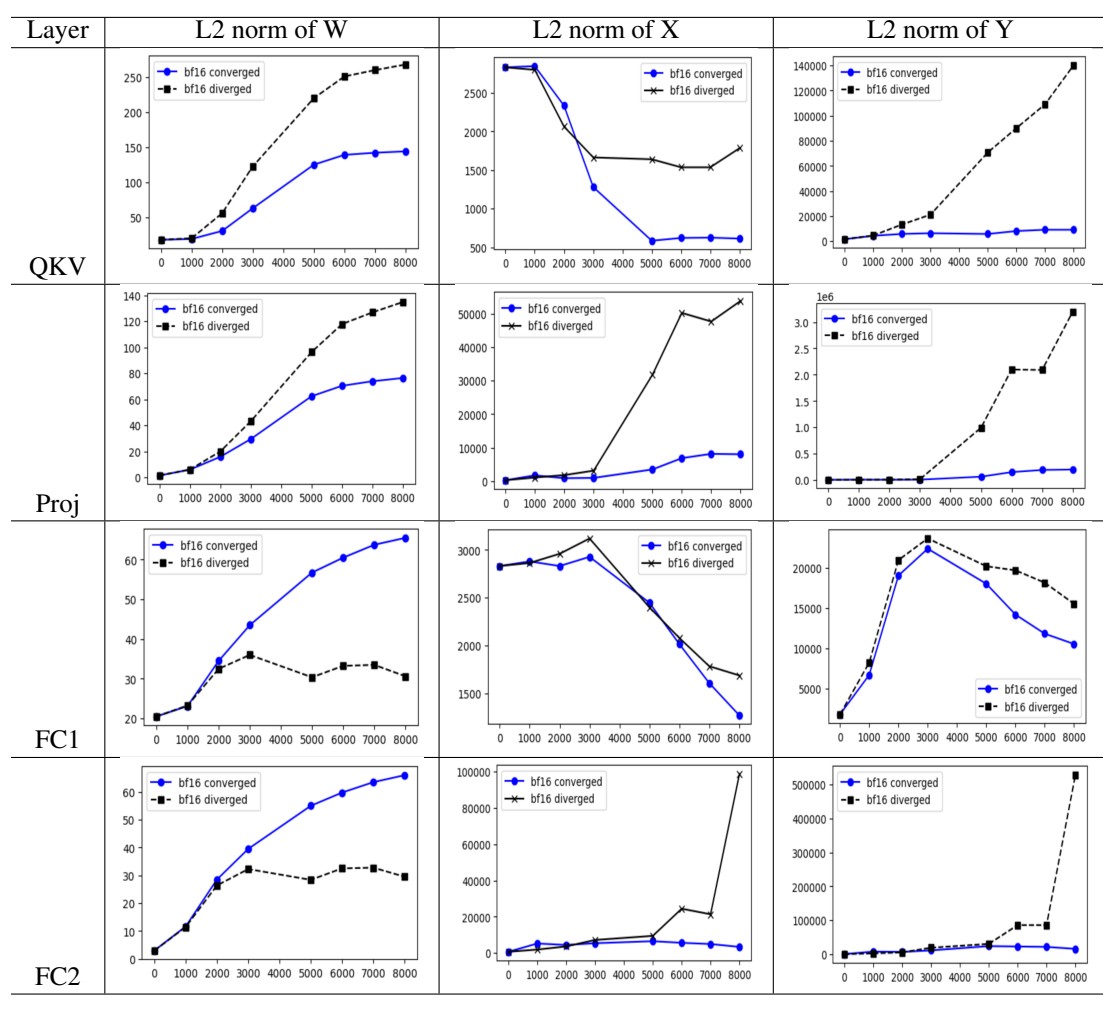

where $\sigma(W) \in \mathbb{R}$ is the spectral norm of $W$ and $\gamma \in \mathbb{R}$ is a learnable parameter, initialized to 1. For more details please refer to Zhai* et al. (2023). This approach influences the magnitude of the linear layer weights and successfully prevents entropy collapse in the attention layers, promoting more stable training. So we apply it on all linear layers of the Transformer block, shown on Figure 1.

## 4.2 SOFTMAX TEMPERATURE (*soft_temp*)

Simple option of controlling logits magnitude (for improving model training stability with high learning rate) is to use softmax temperature. For this method transformer block, shown on Figure 1, computes attention logits and softmax weights as:

$$logit = \tfrac{1}{\sqrt{d}}(XW^Q)((XW^K))^T, \qquad (2)$$

$$softmax[\beta * logit],$$

where $\beta$ is is the temperature of the softmax function Hinton et al. (2015); $d$ is query/key dimension, $X$ is the input, $W^Q$ is the query weight matrix, and $W^K$ is the key weight matrix; and *logit* is the output of block *BMM1* on Figure 1. In our experiments we use $\beta = 0.5$ and label this approach as *soft_temp*.

Table 2: L2 norm of gradient in QKV and FC1 layers (depending on training iteration).

| L2 of *QKV* gradient | L2 of *FC1* gradient |
|---|---|

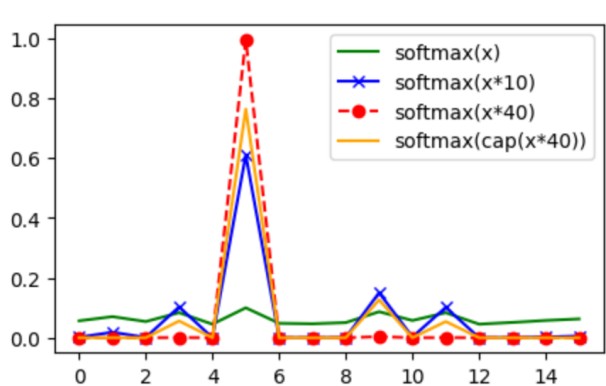

Figure 3: Softmax outputs.

### 4.3 SOFTMAX CAPPING (*soft_cap*)

Another option of controlling magnitude of the logits is softmax capping [Bello et al. (2017), Team et al. (2024)]. Transformer block, shown on Figure 1 (block Softmax), will use softmax capping as:

$$capped\_softmax(logit, capping) = softmax[tanh(logit/capping) * capping], \qquad (3)$$

where *capping* is attention logit capping coefficient, *logit* is the output of block *BMM1* on Figure 1 and defined by equation 2. Softmax capping can be interpreted as an adaptive method of softmax temperature control. For example if we apply softmax capping (with *capping*=10) on uniform distribution *x* with magnitude 40 (discussed in section 3 and shown on Figure 3 as red curve), then the output will be an orange curve (shown on Figure 3). As we can see, softmax capping will make the output close to the softmax of *x* with magnitude 10 (blue curve on Figure 3). In our experiments below, we use *capping* = 50 and label this approach as *soft_cap*.

### 4.4 SOFTMAX CLIPPING (*soft_clip*)

In Bondarenko et al. (2023) authors propose to replace softmax function on Figure 1 with the following clipped softmax:

$$clipped\_softmax(logit; \zeta, \gamma) = clip[(\zeta - \gamma) \cdot \text{softmax}(logit) + \gamma, 0, 1]$$

where $\zeta \geq 1$, $\gamma \leq 0$ are hyper-parameters of the method; *logit* is the output of block *BMM1* on Figure 1 and defined by equation 2. In this experiment we use $\zeta$=1.03 and $\gamma$=-0.03. This approach can improve model training stability: whenever softmax values are clipped they will not give a gradient, preventing the outliers from growing further Bondarenko et al. (2023). We label this method as *soft_clip*.

## 4.5 LAYERSCALE

*LayerScale* Touvron et al. (2021) adds learnable feature scaling after every residual block. It does a per-channel multiplication of the vector produced by each residual block, as shown on Figure 4. It is a method of feature scaling: it does per channel multiplication of features with learnable parameters. For more details please refer to Touvron et al. (2021).

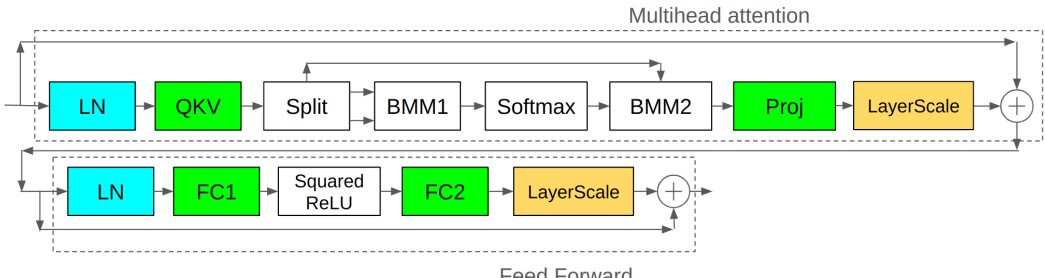

Figure 4: LayerScale in Transformer block.

## 4.6 QK LAYER NORMALIZATION (*QK_norm*)

In Dehghani et al. (2023) authors observe divergence training loss due to large values in attention logits, which leads to (almost on-hot) attention weights in the output of the softmax. To address the model divergence, [Dehghani et al. (2023), Henry et al. (2020)] propose to use layer normalization to the queries and keys before the dot-product attention computation, as shown on Figure 5. We label this method as *QK_norm*.

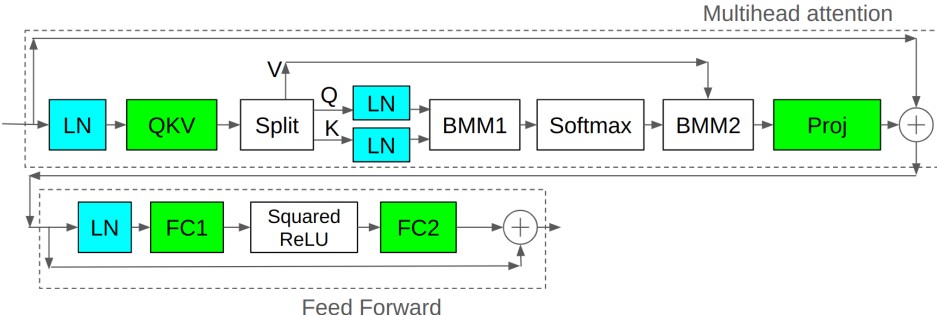

Figure 5: QK layer normalization in Transformer block.

## 4.7 COMBINATION OF QK LAYER NORMALIZATION WITH SOFTMAX CAPPING (*QK_norm_cap*)

QK layer normalization (discussed in section 4.6) controls magnitudes of Q and K features before the dot product (computed in BMM1 block) and softmax. It improves model training stability as shown in [Henry et al. (2020); Dehghani et al. (2023); Wortsman et al. (2024)]. Softmax capping, discussed in section 4.3, controls the softmax temperature which also can be helpful for reducing softmax sensitivity to large magnitude of input logits. So we hypothesize that combination of QK layer normalization with softmax capping can compliment each other and further improve model stability. That is why we combine both of these options as follows:

$$logit = \frac{1}{\sqrt{d}}\mathrm{LN}(XW^Q)(\mathrm{LN}(XW^K))^T,$$

$$capped\_softmax(logit, capping) = softmax[tanh(logit/capping) * capping],$$

where LN stands for layer normalization, *logit* is the output of block *BMM1* on Figure 5; *capping* is described in section 4.3. We label this method as *QK_norm_cap*.

## 4.8 LAYER NORMALIZATION AFTER QKV LAYERS (*QKV_norm*)

In the model with *QK* layer normalization (discussed in section 4.6) we apply layer normalization before and after *QKV* linear layer. Given that we observe magnitude explosion in the output of *QKV* layer, we hypothesize that layer normalization after *QKV* layer should address the issue and there is no need to apply layer normalization before *QKV* layer, This approach is shown on Figure 6, we labeled it as *QKV_norm*. Similar idea was proposed in Menary et al. (2024).

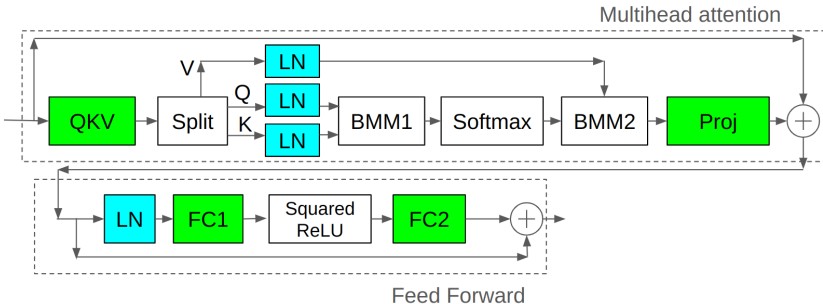

Figure 6: Transformer block with added layer normalization after QKV layer (and removed LN before QKV).

## 4.9 LAYER NORMALIZATION AFTER QK, PROJ AND FC2 LAYERS (*QK_FC_norm*)

In section 3 we observe that the magnitude of all linear layers of the diverging model is much higher in comparison to the magnitude of the converging model. Particularly in layers: *QKV*, *Proj* and *FC2*. It prompts us to apply layer normalization after the QK (as it was done in Dehghani et al. (2023) and in addition use layer normalization after *Proj* and *FC2* layers as shown on Figure 7. We label this method as *QK_FC_norm*. Gemma2 Team et al. (2024) has similar topology (it applies pre and post normalization on both attention and feed forward modules), but it does not have QK layer normalization.

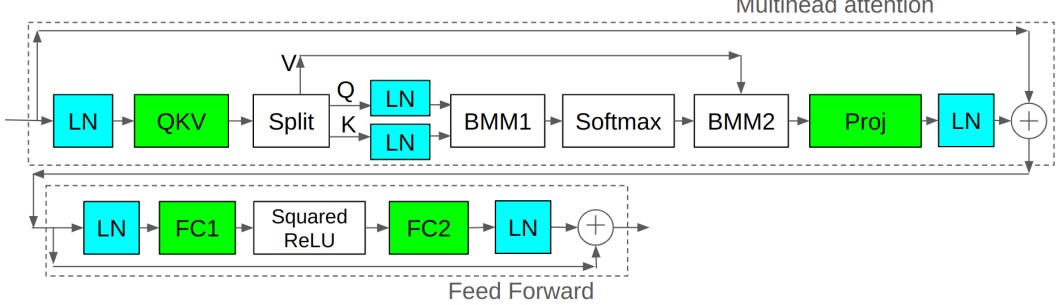

Figure 7: Transformer block with added layer normalization after QKV, FC2 and Proj layers.

## 5 EXPERIMENTAL RESULTS

As in Wortsman et al. (2024) we consider a model to be more stable if it can be trained with higher learning rate without divergence. So we train a baseline bf16 model (shown in section 2) and models presented in section 4, with different learning rates: 6e-3; 8e-3; 20e-3; 40e-3; 60e-3 and 80e-3 (with the same initialization seed value). In Table 3 we report whether a model converges or diverges

Table 3: Models divergence/convergence depending on learning rate.

| Method/LR | 6e-3 | 8e-3 | 20e-3 | 40e-3 | 60e-3 | 80e-3 |
|---|---|---|---|---|---|---|
| bf16 baseline | ✓ | x | x | x | x | x |
| *soft_temp* | ✓ | ✓ | x | x | x | x |
| *soft_clip* | ✓ | ✓ | x | x | x | x |
| $\sigma$Reparam | ✓ | ✓ | ✓ | x | x | x |
| *LayerScale* | ✓ | ✓ | ✓ | x | x | x |
| *soft_cap* | ✓ | ✓ | ✓ | ✓ | x | x |
| *QK_norm* | ✓ | ✓ | ✓ | ✓ | x | x |
| *QK_FC_norm* | ✓ | ✓ | ✓ | ✓ | x | x |
| *QKV_norm* | ✓ | ✓ | ✓ | ✓ | ✓ | x |
| *QK_norm_cap* | ✓ | ✓ | ✓ | ✓ | ✓ | x |

Table 4: Models perplexity with confidence interval $\pm0.1$ at 95% level.

| bf16 baseline | *soft_cap* | *QKV_norm* | *QK_norm_cap* | *QK_norm* | *QK_FC_norm* |
|---|---|---|---|---|---|
| 11.19 | 11.24 | 10.85 | 11.00 | 10.84 | 10.87 |

(by checking validation loss function as described in section 3). We label model convergence and divergence by '✓' and 'x' accordingly.

First we present results of bf16 baseline with baseline methods of improving model training stability: $\sigma$Reparam (section 4.1), *soft_temp* (section 4.2), *soft_cap* (section 4.3), *soft_clip* (section 4.4), *LayerScale* (section 4.5) and *QK_norm* (section 4.6). In Table 3 we show that *soft_cap* and *soft_clip* diverge at learning rate 20e-3. The $\sigma$Reparam and *LayerScale* diverge at learning rate 40e-3. Most stable baseline models are *soft_cap* and *QK_norm* which diverge at learning 60e-3.

Layer normalization after QK, Proj and FC2 layers in method *QK_FC_norm* (presented in section 4.9) does not improve model stability in comparison to *QK_norm* and *soft_cap*. Even though *QK_FC_norm* normalizes the outputs of all linear layers, where we observed output magnitude explosion (discussed in section 3) it makes no difference in comparison to *QK_norm* method. It suggests that the main reason for divergence is in QK layers. This observation is aligned with Dehghani et al. (2023). That is why in the experiments below we are focused on QKV layer only.

Our results show that *QK_FC_norm* method did not improve model training stability (in comparison to *QK_norm*). Instead, we hypothesize that combining *QK_norm* with a non-layer-norm approach can improve model stability: these techniques are addressing logits growth at different computation stages and potentially can complement each other. In this paper we combined *QK_norm* with soft capping *soft_cap* in *QK_norm_cap* model and observe that learning rate can be increased by 1.5x without model divergence in comparison to the baseline *QK_norm* Dehghani et al. (2023).

We hypothesize that application of two layer norm before and after the QKV layer in *QK_norm* method (shown in section 4.6) does not bring much value, so we propose to remove layer norm before *QKV* layer and add layer norm after *QKV* as discussed in section 4.8. This approach *QKV_norm* allows us to increase learning rate by 1.5x (without model divergence) in comparison to a method based on *QK_norm*.

We select a set of models: *soft_cap*, *QKV_norm*, *QK_norm_cap*, *QK_norm*, *QK_FC_norm* (which converge with learning rate 40e-3) and train them on 0.2T tokens with normal learning rate 3.0e-4. So that we can estimate the impact of model stability improvements on its accuracy (we use perplexity as accuracy metric). Perplexity of the models are presented in Table 4. We show that model with *soft_cap* has no significant perplexity difference with the bf16 baseline model. We observe significant perplexity improvements with *QKV_norm*, *QK_norm_cap*, *QK_norm*, *QK_FC_norm* models in comparison to the bf16 baseline model.

## 6 CONCLUSION

We did thorough analysis of linear layers in a transformer block (when LLM was diverging) and demonstrated that input activations and outputs of linear layers of a diverging model have much higher L2 norms in comparison to a converging one. We also observed that *QKV*, *Proj* and *FC2* layers have the largest output magnitude. This prompted us to apply layer normalization not only after *QK* layers (as it was done in [Dehghani et al. (2023), Wortsman et al. (2024)]) but after *Proj* and *FC2* layers too in *QK_FC_norm* model, however this did not improve model stability in comparison to *QK_norm* method. This result led us to be focused on *QK* and *QKV* layers only. So we proposed to combine *QK_norm* with soft capping *soft_cap*: these techniques are addressing logits growth at different stages. We hypothesized that their combination(*QK_norm_cap*) can further improve model stability by complementing each other. We observed that with *QK_norm_cap*, learning rate can be increased by 1.5x (without model divergence) in comparison to a *QK_norm* Dehghani et al. (2023).

We hypothesized that application of two layer norm before and after the QKV layer in *QK_norm* method (shown in section 4.6) does not bring much value: output of QK layers are already normalized by QK layer norm and there is no need to also normalize the input of QKV layer (as it is done in section 4.6). So we proposed to remove layer norm before *QKV* layer and add layer norm after *QKV* as discussed in section 4.8. We observed that this approach *QKV_norm* allowed us to increase learning rate by 1.5x (without model divergence) in comparison to *QK_norm* model.

We estimated perplexity of the most stable models, listed in Table 4 using normal training mode with learning rate 3e-4. We observed significant perplexity improvements with *QKV_norm*, *QK_norm_cap*, *QK_norm*, *QK_FC_norm* models in comparison to the bf16 baseline model.

The methods presented in this work improved model training stability on small language models (830M parameters) with significant improvements in perplexity. A future focus is testing these approaches on much larger models with more tokens and getting benchmark results for our new model architectures.

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
