# OpenReview forum: "METHODS OF IMPROVING LLM TRAINING STABILITY"
_ICLR.cc/2025/Conference — ICLR 2025 Conference Withdrawn Submission_

### Official Review · Reviewer_hpZH · 2024-10-18

**Soundness:** 2
**Presentation:** 2
**Contribution:** 2
**Rating:** 3
**Confidence:** 3

**Summary:**

Based on the analysis of the L2-Norm the authors experimented with the addition and removal o fLayer-Normalizations within the classical transformer block architecture. In combination with Softmax-capping the authors achieve stability with significantly increased learning rates alongside perplexity improvements relative to a variety of baselines.

**Strengths:**

* Improvements seem significant in terms of perplexity
* Good number of comparisons

**Weaknesses:**

* Given the availability of H100 GPUs to the authors investigating the properties at fp8 would have been interesting too, as with lower precision comes greater instability. This would significantly raise the relevance of the publication.
* The datasets were not disclosed making reproducibility virtually impossible. While this is sadly a common practice for a lot of LLM paper, given the small parameter count of the model I feel reproducing the experiments would be more realistic, so I do not see why those were not disclosed.
* I feel the training time with with 1T tokens per runs is excessivelz long given the size of the model. A chinchilla-optimal [Hoffmann et. al. 2022] training regime would be closer to 16B-tokens, which seems more adequate for this type of work.
* There is a severe lack of ablation studies in various directions that would help this paper to effectively increase the significance of their findings. Interesting selling points here would have been: "How much training time could be saved by increasing the learning rate" and more general "How is convergence (besides the binary distinction of convergence and divergence) impacted? How does scaling the model architecture impact the findings?' The latter I find particularly important as a single training of a 3B-parameter models could also provide additional metrics on known benchmarks to give us more insight into the results that go beyond the fairly uninformative perplexity score.
* The writing needs work and I found several typos in the work, but this can be corrected easily.

**Questions:**

* Did you test this architecture with a larger parameterization like 2-3B parameters?
* Please disclose the training datasets and the mixture ratios?
* If one looks to maintain a certain perplexity-level in terms of predictive performance, how much compute can be saved by increasing the learning rate and by extension shortening the training?
* How will fp8-based setups impact your findings? Is this approach viable with fp8?
* Why was such a long training time chosen when chinchilla-optimal training would be significantly less and would open up more resources for ablation studies?

---

### Official Review · Reviewer_Bmcd · 2024-11-02

**Soundness:** 3
**Presentation:** 3
**Contribution:** 2
**Rating:** 5
**Confidence:** 3

**Summary:**

This paper extends prior work on training stability to analyze when higher learning rates can affect model divergence. It looks at how increasing the learning rate can increase the L2 norm of all linear layers in a Transformer block and cause the model to diverge. The paper then compares layer normalization methods to mitigate this divergence with a 830M parameter model. It shows how learning rate can be increased without causing model divergence, while improving perplexity.

**Strengths:**

-The paper does a good job of comparing and contrasting a range of methods to mitigate model divergence when training.

-The paper reports which methods worked, as well as which did not offer any benefit. Both of these results are incredibly valuable to analyzing model divergence as they offer further intuition for why certain methods do better.

**Weaknesses:**

-The paper should report results on a range of small model sizes. At the moment, there are only results for a 830M parameter model, and it is not obvious how these findings would scale down.

-It would be helpful to provide further motivation and examples for why training instability might be a serious issue that requires mitigation

-Many of the Figures need labels on the axes

-Figure 2: It would be good to have more examples of learning rate to show exactly how the model stops converging

**Questions:**

-Which datasets are used for training?

---

### Official Review · Reviewer_5UGZ · 2024-11-05

**Soundness:** 2
**Presentation:** 1
**Contribution:** 2
**Rating:** 3
**Confidence:** 4

**Summary:**

This paper explores training stability in large language models (LLMs) by examining the impact of high learning rates on model divergence using a smaller 830M parameter model. The authors discover that high learning rates cause the L2 norm of outputs from specific Transformer block layers (QKV, Proj, and FC2) to increase, leading to instability. Their experiments reveal that while extending layer normalization to additional layers does not stabilize the models, combining QK layer normalization with a softmax capping technique successfully prevents divergence, allowing for increased learning rates. They suggest optimizing layer normalization placement for further stability improvements. Their findings show significant perplexity reductions across all models using these methods, compared to a baseline, highlighting the techniques’ efficacy in stabilizing LLM training at higher learning rates and suggesting potential applications in larger, more complex models.

**Strengths:**

This paper proposes new methods of applying layer normalization and softmax capping across various layers in Transformer blocks to enhance training stability and improve perplexity in large language models.

**Weaknesses:**

1.	The paper’s format does not adhere to the ICLR25 standard.
2.	The paper addresses the training stability of large language models (LLMs), stating that larger models tend to have decreased stability. However, the experimental results do not provide evidence that the method can be scaled to models with 10 billion or 100 billion parameters.
3.	The experiments and results lack any comparison of training losses. Instead, the author only provides a table of divergence/convergence rates, which is insufficient to convince me of the method’s validity.
4.	The lack of experimental results, including only training loss convergence and perplexity, is inadequate for evaluating an LLM.
5.	I am also curious about whether the method is effective with Mixture of Experts (MoE), beyond traditional dense LLMs.

**Questions:**

1.	Please consider the aforementioned weaknesses.
2.	There are multiple typos, for example, “The model is trained on a subset of a 1T token dataset with batch size 512 and sequence length 4096 using 32 H100 GPUs H10” in section 2.

---

### Official Review · Reviewer_zbs4 · 2024-11-09

**Soundness:** 2
**Presentation:** 2
**Contribution:** 2
**Rating:** 3
**Confidence:** 3

**Summary:**

This paper studies LLMs training stability with a smaller language model of size 830M parameters and higher learning rate. They found that the reason of the training divergence stems from the growth of the L2 norm of all outputs in QKV, Proj and FC2 linear layers. They also proposed to use layer norm after these layers to improve the training stability.

**Strengths:**

1) The finding is interesting that multiple layers are responsible for training instability because of the output growth.
2) They highlighted very well different training stability methods proposed in the literatures.

**Weaknesses:**

1) Quality of the figures are low, the authors should use vector images.
2) Results section needs more evidence of different experiments they did rather than just two table they showed. The authors could add the training curves of how different methods are helping to improve stability.
3) They study the instability with higher learning rate and make the model training stable for the higher learning rate. There is no evidence if the model is giving training convergence faster at different scales. Since it is mostly an empirical work without any theoretical contribution, I think this is an important study to validate this as an useful proposal.
4) The authors should elaborate the captions of the figures e.g. Figure 3.
5) Overall the paper requires more works both in terms of writing and contributions to validate and explain the proposed idea.

**Questions:**

1) Why did the authors train bf16 model not bf32? Is there any impact of precision on the training stability?
2) Why did the authors change the format/style of the review paper? It is not showing the line number.

---

### Note · Authors · 2024-11-14

**Comment:**

Thank you all for valuable feedback, but given that reviewers are interested in results on:
* Train additional models with 10 billion or 100 billion parameters
* Train additional model with Mixture of Experts (MoE), beyond traditional dense LLMs.
* Train more models with high learning rate and show that proposed approach is speeding up model convergence and can save compute resources.
* Train more models to show that proposed approach scales on different model sizes.
* In addition to bf16 train models with fp8.

It is not feasible to address all of these by the due date, so withdrawing the paper.

**Withdrawal Confirmation:**

I have read and agree with the venue's withdrawal policy on behalf of myself and my co-authors.